# Targeting Cancer Stem Cells: A Strategy for Effective Eradication of Cancer

**DOI:** 10.3390/cancers11050732

**Published:** 2019-05-27

**Authors:** Masahiro Shibata, Mohammad Obaidul Hoque

**Affiliations:** 1Department of Otolaryngology-Head and Neck Surgery, Johns Hopkins University School of Medicine, Baltimore, MD 21231 USA; m-shibata@med.nagoya-u.ac.jp; 2Department of Breast and Endocrine Surgery (Surgery II), Nagoya University Graduate School of Medicine, Nagoya 4668550, Japan; 3Department of Oncology, Johns Hopkins University School of Medicine, Baltimore, MD 21231, USA; 4Department of Urology, Johns Hopkins University School of Medicine, Baltimore, MD 21231, USA

**Keywords:** cancer, cancer stem cell, YAP1, verteporfin, combinational therapy

## Abstract

Cancer stem cells (CSCs) are subpopulations of tumor cells with the ability to self-renew, differentiate, and initiate and maintain tumor growth, and they are considered to be the main drivers of intra- and inter-tumoral heterogeneity. While conventional chemotherapy can eradicate the majority of non-CSC tumor cells, CSCs are often drug-resistant, leading to tumor recurrence and metastasis. The heterogeneity of CSCs is the main challenge in developing CSC-targeting therapy; therefore, we and other investigators have focused on developing novel therapeutic strategies that combine conventional chemotherapy with inhibitors of CSC-regulating pathways. Encouraging preclinical findings have suggested that CSC pathway blockade can indeed enhance cellular sensitivity to non-targeted conventional therapy, and this work has led to several ongoing clinical trials of CSC pathway inhibitors. Our studies in bladder cancer and lung adenocarcinoma have demonstrated a crucial role of YAP1, a transcriptional regulator of genes that promote cell survival and proliferation, in regulating CSC phenotypes. Moreover, using cell lines and patient-derived xenograft models, we showed that inhibition of YAP1 enhances the efficacy of conventional therapies by attenuating CSC stemness features. In this review, we summarize the therapeutic strategies for targeting CSCs in several cancers and discuss the potential and challenges of the approach.

## 1. Introduction

The aggressive nature of certain solid tumors is thought to result, at least in part, from the expansion of cancer stem cell (CSC) populations with self-renewal and differentiation properties that ultimately lead to intra- and inter-tumoral heterogeneity [1,2,3,4,5]. Indeed, the ratio of CSCs to non-CSCs in a tumor cell population correlates with poor clinical outcome in many cancers [6]. While chemotherapy usually eradicates the bulk population of non-CSC tumor cells, CSCs are relatively chemoresistant and not only become enriched after chemotherapy [7,8,9,10,11,12,13] but also enter a transient dormant status that may result in tumor recurrence and metastasis [14,15]. Recent work suggests that CSCs may contribute to therapy resistance by repopulating residual tumors between chemotherapy cycles [14]. Thus, a promising and urgently needed therapeutic strategy for many solid tumors is to develop novel therapies that inhibit specific CSC-generating and -expanding pathways in a given tumor [16,17].

Increasing evidence suggests that blockade of factors and/or pathways necessary for CSC survival and growth can increase the cellular sensitivity to conventional chemotherapy, and several clinical trials aiming to abrogate CSCs are currently ongoing [4]. However, it is important to note that CSCs are not a single cell type but instead are composed of multiple heterogeneous phenotypes, making it extremely difficult to predict whether a specific CSC-targeting therapy will be effective for an individual patient. Furthermore, there are currently no known universal markers for CSCs, which eliminates the possibility of developing a ‘pan-CSC’-targeting therapy [18,19]. Due to this diversity among CSCs, there is a need to identify CSC-specific markers or regulatory pathways characteristic of a given tumor to enable the development of tailored therapies. In other words, effective targeting of CSCs will require a better understanding of the intra- and inter-tumoral diversities of CSC types and functions.

In this regard, the past several years have seen increasing efforts to develop CSC-targeting therapeutic approaches. One of these is YAP1, a transcriptional regulator and the main effector of the Hippo pathway, which normally regulates cellular proliferation and differentiation but also promotes malignant phenotypes and drug resistance in various cancers [20]. Our group recently demonstrated that YAP1 regulates the generation and expansion of CSCs in urothelial carcinoma of bladder and lung adenocarcinoma and that pharmacological inhibition of YAP1, in combination with chemotherapies, suppresses cancer progression by attenuating cancer stemness features [20,21,22].

Here, we review current knowledge of some of the potential mechanisms by which CSCs can be targeted and the therapeutic challenges faced in eliminating CSCs in various cancers. We also discuss our recent works on CSCs with a particular focus on YAP1-targeting strategies.

## 2. CSC-Targeting Therapies

Recently, several attempts have been conducted aiming to combine conventional drugs with CSC-targeting agents (Figure 1) [6,19]. However, the heterogeneity and high plasticity of not only CSCs but also many non-CSC tumor cells amplify the challenge of identifying suitable targeting molecules for diverse cancers [23]. Such plasticity enables non-CSCs to transdifferentiate into CSCs that may be resistant to specific CSC-targeting therapies [24]. In addition, many surface markers are also expressed on normal stem cell populations where they may play essential roles in tissue hemostasis and renewal [24]. Therefore, there is an urgent need to identify novel CSC specific molecules and/or pathways for specific cancer types that are essentially absent (or expressed at low levels) in normal stem cells.

Numerous studies have described the existence of CSC populations in a variety of cancers, including breast, colorectal, lung, ovary, bladder, and head and neck cancers. Malignant stemness markers and factors differ among cancer types, among the same cancer in different patients, and among different regions within the same tumors; nevertheless, all CSCs share the ability to drive tumorigenesis, metastasis, drug resistance, and immunosuppressive microenvironment [6,25]. These features make CSCs attractive targets for the development of more effective treatments for cancer [6]. To date, attempts have been made to target stemness markers, such as STAT3 and NANOG [6], as well as pathways that regulate malignant stemness, most notably the Wnt/β-catenin, Notch, hedgehog, and JAK-STAT pathways, which promote stemness features in various cancers [26]. We summarize these attempts and current status in Table 1. In the following sections, we describe some of the CSC markers and pathways that are subject to recent efforts to develop therapeutic strategies in several major cancers.

### 2.1. Breast Cancer

CSCs in breast cancer were first identified as cells with a CD44^+^CD24^−/low^ phenotype [53]. In this pioneering study, putative CSCs and non-CSCs were tested for their ability to form tumors in mice. Transfer of as few as 100 CD44^+^CD24^−/low^ cells resulted in tumor formation, whereas the transfer of thousands of other cells with various phenotypes did not [53]. Subsequent studies have identified CD44, CD133, and ALDH as CSC markers in breast cancer [54,55]. In addition, several transcription factors, such as KLF4, NANOG, OCT4, and SOX2, have been reported to be expressed and induce stemness features in breast CSCs [56,57,58]. Interestingly, while breast cancer sphere-forming cells showed high SOX2 expression, NANOG and OCT4 were not expressed in these cells, suggesting the presence of heterogeneous CSC populations [57]. In addition, several deregulated signaling pathways, including hedgehog, Notch, TGF-β/BMP, and Wnt/β-catenin are known to contribute to the enhancement of characteristic CSC phenotypes, such as self-renewal, proliferation, apoptosis evasion, invasion, epithelial-mesenchymal transition (EMT), and drug resistance [59,60].

To date, several approaches to therapeutic targeting of CSC populations have been investigated in breast cancer. The percentage of CD44^+^CD24^−^ cells is higher in basal-type compared with luminal-type breast cancer [61]; therefore, attempts to target CSCs have mainly focused on the basal subtype. Inhibitors of histone deacetylases, which are important epigenetic modifying enzymes, reduced the abundance of CD44^+^CD24^−/low^-expressing cells, inhibit the activity of ALDH1, and suppress the expressions of several stemness markers, such as *BMI1*, *NANOG* and *OCT4* [27,28]. Recently, treatment of breast CSCs with interferon-β in vitro has been reported to limit stemness, migration, sphere-forming properties, and re-expression of CD24, and promote an epithelial-like morphology [62]. Another study found that the type 2 diabetes drug metformin suppresses CSC growth by targeting KLF5 for degradation and preventing transcription of its downstream target genes, *NANOG* and *FGF-BP1* [37]. Interestingly, metformin has been mentioned as a potential CSC-targeting drug for use as (neo-)adjuvant therapy [38]. The possibility that cytotoxic drugs can selectively be delivered to CSCs is supported by the demonstration that iron oxide magnetic nanoparticles containing anti-CD44 antibody and gemcitabine derivatives can specifically target and kill CD44^+^ cells [33].

Several therapeutic agents have been evaluated to target the Wnt/β-catenin signaling pathway, which is an important regulator of CSC characteristics. Resveratrol, a natural polyphenolic compound, reduced the breast CSC population in mice via inhibition of Wnt/β-catenin signaling [47], and a highly potent small molecule antagonist of β-catenin binding to nuclear T-cell factor has been shown to inhibit the growth of breast CSCs and, to a lesser extent, non-CSCs [63]. Pyrvinium pamoate, an anthelmintic drug and inhibitor of the Wnt/β-catenin pathway, prevented the proliferation of breast cancer cells, especially CD44^+^CD24^−/low^ and ALDH^+^ CSCs, via downregulating NANOG, OCT4, and SOX2 [46]. In a recent study, carboplatin treatment activates STAT3, leading to breast CSCs enrichment, and combination treatment with a STAT3 inhibitor and carboplatin attenuated the stemness-like features, resulting in a more efficient therapeutic response [15].

Micro RNAs (miRNAs) regulate gene expression by destabilizing and/or silencing the translocation of target mRNAs, and many miRNAs with CSC-promoting or -suppressing properties have been investigated as potential therapeutic targets. For example, the tumor suppressor miR-223 is downregulated in CD44^+^CD24^−/low^ triple-negative breast CSCs, and its overexpression resensitized the cells to induction of apoptosis [64]. Inhibition of miR-125a, which regulates TAZ, an effector in the Hippo pathway, led to a significant reduction in the breast CSC pool [65], and miR-34a has also been reported to suppress breast CSC-like characteristics by inhibiting the Notch1 signaling pathway [66].

### 2.2. Colorectal Cancer

CD133 has been identified as a marker of colon CSCs, which make up approximately 2.5% of colorectal cancer tumor cells. Notably, CD133^+^ cancer cells rapidly formed tumors after injection into immunodeficient mice, whereas CD133^−^ cells did not [67]. Moreover, CD133^+^ colorectal cancer cells have also been shown to be resistant to radiotherapy and chemotherapy [68], consistent with a CSC phenotype. Another known CSC marker, CD44 is enriched on CSC cells with CSC-like properties and may promote their function by forming a positive feedback loop with Ras signaling [69], and CD26^+^ colorectal CSCs contribute to tumor initiation by facilitating the EMT [70]. The G protein-coupled receptor LGR5 has also been reported to be a marker for colorectal CSCs during the initial stages of tumorigenesis, and *LGR5* expression levels correlated with aggressive clinicopathological features in colorectal cancer [71,72,73]. Interestingly, combination targeting of both LGR5^+^ cells and differentiated cancer cells prevented tumor resistance and relapse [71,72]. Consistent with their roles in other cancer types, the transcription factors NANOG, OCT4, and SOX2 promote stemness features in colorectal CSCs [68,74].

The transcription factor STAT3 is activated by many signaling pathways involved in the regulation of cell growth and apoptosis. Accordingly, STAT3 is an oncogenic driver and contributes to carcinogenesis by promoting cell survival, angiogenesis and the generation and expansion of CSCs, which leads to drug resistance [75,76,77,78,79]. Although further studies are needed, STAT3 is considered to be a promising CSC target in colon cancer [6]. Other study showed that napabucasin, which inhibits STAT3-driven gene transcription, blocks several key molecules in CSC-related signaling pathways, including *c-Myc*, *Nanog* and *Sox2*, in a colon cancer mouse model, thereby suppressing metastases [43].

In addition to being the subject of numerous preclinical studies, several colorectal CSC targeting therapies have advanced through clinical trials. For example, napabucasin could be safely combined with FOLFIRI (folinic acid, fluorouracil, and irinotecan), a standard chemotherapy regimen for colorectal cancer, and showed encouraging signs of efficacy in phase I and II clinical trials [80]. Subsequently, a phase III trial of napabucasin showed prolonged survival of phosphorylated STAT3 positive patients in advanced colon cancer [44].

### 2.3. Lung Cancer

Multiple markers of lung CSCs have been identified, including CD44, CD133, CD166, and ALDH1 [7,81,82]. However, the intra-tumoral heterogeneity and high plasticity of lung cancer have made it difficult to precisely define which markers drive the CSC properties in a given tumor [82]. Among these molecules, CD44, which regulates adhesion, differentiation, homing and migration, is considered to be a key CSC marker [82]. CD44^+^ non-small cell lung cancer (NSCLC) cells were shown to specifically form spheroids and induce tumor initiation via induction of NANOG, OCT4, and SOX2 overexpression [81]. Another study successfully identified lung CSCs as a CD44^+^CD90^+^ subpopulation that displayed a mesenchymal morphology and elevated expressions of *NANOG*, *OCT4*, and EMT-related genes [83]. CD133 has also been identified as a lung CSC marker; notably, CD133^+^ cells have sphere-forming and tumorigenic capabilities [84], are chemoresistant, and are a biomarker of poor prognosis in lung cancer patients [82]. ALDH1^+^ cells also exhibit CSC properties that include the ability to self-renew, initiate tumors, and resist drug therapy [85]. This study also demonstrated that ALDH1^+^ cells co-express CD133 in lung cancer patients, and that ALDH1 expression correlated positively with tumor stage and grade and with poor prognosis [85]. One particularly interesting study has explored the expressions of multiple putative stemness markers, such as ALDH1, CD24, CD44, CD133, OCT4, and SOX2, in 371 surgically resected NSCLC specimens [86]. ALDH1, ABCG5, CD44, and SOX2 were shown to be associated with poor differentiation and increased proliferation. In addition, ALDH1, CD44, and SOX2 expressions were more frequently associated with squamous cell carcinoma whereas CD24 and CD166 were more common in adenocarcinoma, which supports the heterogeneity of stemness markers among CSCs of different histologic types [86]. Like other cancer types, lung CSCs have been reported to utilize common signaling pathways, including the Wnt/β-catenin, hedgehog, and Notch pathways [87].

Among the CSC-targeting strategies explored in lung cancer to date, genetic knockdown or pharmacological inhibition of ALDH1 has been shown to attenuate the proliferative and migratory capabilities of NSCLC cells [88]. Similarly, disulfiram, an ALDH1 inhibitor, showed synergistic effects with copper in attenuating lung CSC activity via downregulation of stemness-related genes [29]. In a study of isolated ALDH1^+^ lung cancer cells with stem cell-like characteristics, combination therapy with diethylamino-benzaldehyde and disulfiram was able to resensitize resistant cells to the cytotoxic effects of cisplatin [30]. In a phase IIb trial of NSCLC, disulfiram was well-tolerated and prolonged overall survival when combined with cisplatin and vinorelbine [31]. As in colorectal cancer, STAT3 has been investigated as a CSC-regulatory target in lung cancer. In a phase I trial of the STAT3 inhibitor OPB-51602, *EGFR*-mutant NSCLC patients were likely to obtain better response among the patients of various cancer types [45]. GDC-0449, an inhibitor of the hedgehog pathway, has also demonstrated efficacy in both NSCLC and small-cell lung cancer via suppression of stemness-related features [36]. Tarextumab, which blocks Notch2 and Notch3 signaling, reduces the abundance of tumor-initiating cells and the growth of lung cancer [39]. A phase Ib trial of tarextumab showed good tolerability and a dose-dependent anti-tumor effect when combined with etoposide and platinum in small-cell lung cancer patients [40].

### 2.4. Ovarian Cancer

Ovarian cancer is an umbrella term for a broad range of neoplasms of different histopathological types, including serous, endometrioid, clear cell, and mucinous; importantly, these subtypes exhibit different responses to drug treatment [89]. The proportion of CSCs in ovarian cancer varies between 0.1% and 30% depending on the disease type and stage [90]. CD133^+^ ovarian cancer cells have been shown to have a more proliferative phenotype than do CD133^-^ cells [91]. A recent study showed that CD133^+^ CSCs induced CD133^−^ cells to undergo the EMT and to display enhanced metastatic capacity via secretion of CCL5 and activation of NF-κB signaling [92]. Another group examined the expression of CD44 on ovarian CSCs by immunohistochemistry and found that its expression was associated with poor prognosis [93]. CD44 knockdown in CSCs suppressed their proliferation, migration/invasion, and sphere-forming abilities and increased their drug sensitivity [93]. ALDH has been reported to be overexpressed in spheroid ovarian CSCs [94], and ALDH1A1 was found to be a predictive biomarker of the response of patients to chemotherapy [95]. Another study also identified an association between ALDH overexpression and poor prognosis and additionally showed that ALDH1A1 overexpression increased the clonogenic potential, expression of DNA repair proteins, and platinum resistance of ovarian cancer cells [96]. ALDH^+^CD133^+^ ovarian CSCs appear to have greater growth potential than do ALDH^+^CD133^−^ cells; indeed, as few as 11 ALDH^+^CD133^+^ cells were required to form tumors in mice [97]. Another study reported that ovarian CSCs were enriched among cells co-expressing ALDH1 and CD44 and that these cells displayed chemoresistance [98]. The authors proposed that high ALDH1 alone may be insufficient to drive the ovarian CSC phenotype, but that ALDH1 and CD44 co-expression may trigger chemoresistance and poor clinical outcome [98].

Many studies have investigated CSC targeting as a therapeutic strategy for ovarian cancer. For example, miR-199a decreased CD44 expression, resulting in reduced tumorigenicity and increased chemotherapeutic sensitivity [99]. In this study, other stemness factors, such as *NANOG*, *OCT4* and *CD133*, were also reduced by miR-199a overexpression [99]. Targeting of CD133^+^ cells with an anti-CD133 antibody-toxin conjugate was shown to inhibit the progression of ovarian cancer [34]. ALDH1 overexpression in ovarian CSCs increased taxane and platinum resistance, and conversely, *ALDH1* knockdown overcomes the resistance, enhancing growth inhibition in the presence of chemotherapeutic agents [95]. Treatment of ovarian CSCs with solanum incanum extract inhibited ALDH1 activity and Notch1 and FoxM1 expression, leading to attenuation of stemness features and increased sensitivity to cisplatin and paclitaxel [32]. Imatinib, a small molecule inhibitor of c-Kit (CD117), in combination with platinum cytotoxic drugs inhibited ovarian CSC activity by blockade of the Wnt/β-catenin signaling pathway [48]. However, imatinib had only a modest impact on the prognosis of ovarian cancer patients in phase II clinical trials [49,50]. These unfavorable results may have been due to intra- and/or inter-tumoral heterogeneity among both CSCs and non-CSCs.

### 2.5. Bladder Cancer

The existence of bladder CSCs was first reported in 2009 and has since been confirmed in numerous studies [100]. These cells are thought to be a major reason for the failure of adjuvant treatment and poor therapeutic outcomes of bladder cancer [100,101]. Bladder CSCs can be identified using a number of cell surface markers, including CD44, CD67LR, EMA, and ALDH1A1, and the hedgehog, phosphoinositide 3-kinase (PI3K)/AKT, Wnt/β-catenin, and Notch signaling pathways, which play key roles in the generation, expansion, and the maintenance of stemness, self-renewal, and proliferative potential in these heterogeneous groups of cells [102]. Although agents that target a small number of these pathways and markers are available and some of them are being tested in clinical trials, a clear treatment strategy has not been developed.

One study has shown that basal subtype bladder cancer cells expressing sonic hedgehog signaling molecule (SHH), a vertebrate homolog of *Drosophila* hedgehog, are capable of regenerating all cell types within the urothelium [103], suggesting they are CSCs. Subsequent studies suggested that muscle-invasive bladder carcinoma arises exclusively from SHH-expressing stem cells in the basal urothelium [104] and confirmed the role of SHH in the promotion of EMT, tumorigenicity, and stemness [35,105]. Accordingly, targeting of SHH by cyclopamine was shown to inhibit bladder tumor growth [35].

Available data suggest that targeting of the PI3K/AKT pathway inhibits urothelial tumor growth and metastasis by reducing CSCs and blocking the EMT [106]. The main advances in the area of bladder CSCs involve the PI3K/AKT inhibitor myrtucommulone-A and the multi-targeted tyrosine kinase inhibitor motesanib. Iskender et al. found that myrtucommulone-A treatment of the human bladder cancer cell line HTB-9 decreased cell viability, proliferation, and sphere-forming ability and suppressed the expression of common stem cell markers (e.g., CD44, NANOG, OCT4, and SOX2) and other proteins involved in the PI3K/AKT pathway [41]. Similarly, Ho et al. reported that motesanib decreased the expression of survival-related genes in the PI3K/AKT pathway, and combination treatment with motesanib synergistically enhanced the anticancer effect of cisplatin [42].

Elevated activity of the Wnt/β-catenin pathway, which regulates stem cell homeostasis, and increased expression of its components are associated with cancer progression [107]. Among bladder cancer patients, 20 single nucleotide polymorphisms in 40 genes in the Wnt/β-catenin pathway were associated with an increased risk of cancer [108]. A recent study revealed that miR-374a directly targets WNT5A and that low levels of this miRNA were associated with poor prognosis [109]. In this study, miR-374a treatment inhibited the phosphorylation and nuclear translocation of β-catenin, decreasing the expression of cancer stemness-related proteins [109].

The Notch signaling pathway interacts with Wnt and SHH pathways and is involved in the maintenance of stemness features [110,111]. This pathway, which has both oncogenic and tumor-suppressive effects depending on the tissue and cellular context [112], includes four receptors, NOTCH1-4, and five ligands and regulates the transcription of multiple target genes [102]. *NOTCH1* expression is decreased in bladder cancer and its activation reduces cellular proliferation, suggesting that it has a tumor-suppressive role [112]. In contrast, *NOTCH2* acts as an oncogene by promoting cell proliferation, the EMT, and maintenance of stemness; moreover, its inhibition attenuates malignant phenotypes in vitro and in vivo [113]. Due to these contrasting roles and the potential for off-target side effects in other organs, more evidence will be required to support the development of Notch-targeting therapy in cancer [106].

### 2.6. Head and Neck Cancer

As in other cancer types, CD44^+^ cells have been reported to possess the unique properties of CSCs in head and neck squamous carcinoma (HNSCC) [114]. In addition, CD24^+^CD44^+^ cells were found to be more chemoresistant and tumorigenic compared with CD24^−^CD44^+^ cancer cells [115]. CD133 has also been investigated as a CSC marker in HNSCC; consistent with this, CD133^+^ cells showed reduced sensitivity to paclitaxel [116,117]. In another study, CD10^+^ HNSCC cells expressed OCT3/4, were relatively resistant to cisplatin and radiation, and had sphere-forming and aggressive tumorigenic properties [118], suggesting that CD10 may be a marker of HNSCC CSCs. In nasopharyngeal carcinoma, the expression levels of NANOG, OCT4, and SOX2 were positively correlated with EMT-associated indicators and worse prognosis [119]. In an oral squamous carcinoma mouse model, Lin28, a key factor for cellular reprogramming and generation of induced pluripotent stem cells, was found to promote stemness properties via inhibition of Let-7 and induction of Oct4 and Sox2 [120]. ALDH1 has also been considered to be a CSC marker in HNSCC; accordingly, increased ALDH1 activity enhances tumorigenesis and decreases the effects of chemoradiotherapy [121]. Moreover, ALDH^+^CD44^+^ HNSCC cells are more tumorigenic and have stronger stemness characteristics than cells expressing CD44^+^ alone [117].

The oncoprotein BMI1 cooperates with MYC and Snail signaling to suppress cellular senescence and apoptosis and to promote cell proliferation and the EMT [122]. In laryngeal carcinoma, BMI1 was co-expressed with CD133 and found to have proliferation-promoting and anti-apoptotic effects [123]. Examination of specimens from patients with oral lichen planus showed that high BMI1 levels were associated with a higher risk of progressing to oral squamous cell carcinoma [124]. In another study, the expression of BMI1 or ALDH1 was associated with the transformation from oral erythroplakia to cancer [125]. Chemoradiation is a standard therapy for HNSCC, and CSCs generally show resistance to both chemotherapy and radiotherapy via control of cell cycling, efficient DNA repair, elevated free radical scavenging, and microenvironment support [117]. Indeed, in several studies, BMI1 expression was associated with a lack of response to chemoradiotherapy [126,127]. Although there is currently little supportive evidence, it is possible that eliminating CSCs by targeting CSC generating and maintaining pathways may increase the sensitivity of HNSCC to chemotherapy and radiotherapy.

## 3. Targeting CSCs by Inhibiting YAP1

In an attempt to target CSC-regulating effector molecules, we have focused on YAP1, a principal effector of the Hippo signaling pathway that regulates organ growth in normal tissues. Increasing evidence suggests that YAP1 has diverse roles in tumorigenesis and drug resistance [20], and it has been reported to promote malignant features (e.g., cell proliferation, invasion, migration, and anti-apoptosis) [20,128,129,130,131,132] and drug resistance in various cancers, including breast, bladder and lung cancer (Figure 2) [51,52,133]. Recent studies have also shown that YAP1 contributes to the establishment of an immunosuppressive tumor microenvironment [20]. Moreover, there is substantial evidence that YAP1 regulates malignant cell stemness [21]. YAP1 is capable of reprogramming non-CSCs into cells that have CSC-like characteristics [134] and helps cancer cells to maintain their stemness by promoting autophagy [135,136]. In NSCLC, YAP1 directly interacts with OCT4 followed by SOX2 upregulation to facilitate self-renewal and acquisition of endothelial-like properties (vascular mimicry) of CSCs [137]. In ovarian cancer, YAP1 regulates pluripotency and chemoresistance [138]. YAP1 induction concurrently induces two potential stemness markers, EPCAM and keratin 19 in hepatocellular carcinoma [139]. Most recently, YAP1 has been reported to activate SOX9 to confer CSC-like features in esophageal squamous cell carcinoma [140]. In addition, YAP1 signaling is activated by long noncoding RNAs in liver cancer, leading to the induction of self-renewal by CSCs [141]. Importantly, YAP1 interacts with numerous oncogenic signaling pathways, including MAPK, PI3K/mTOR, and Hippo, which enhance tumorigenesis [142]. The interplay between YAP1 and the Wnt/β-catenin pathway is known to be important for the maintenance and expansion of CSCs [142,143]. For example, in the absence of Wnt signaling, YAP1 is restricted to the β-catenin destruction complex, but activation of Wnt signaling induces their release and translocation to the nucleus, where they upregulate transcription of β-catenin and YAP1 target genes [144]. Cancerous tissue is composed of not only cancer cells but also mesenchymal cells, endothelial cells, and immune cells [20]. Interestingly, YAP1 in these cells also contributes to cancer progression. For example, in a stromal cell, YAP1 upregulates transcription of myofibroblast marker, such as *CYR-61* and *CTGF* for activating cancer-associated fibroblast that is required for remodeling cancer stroma [145]. In naïve T-cell, YAP1 promotes polarization to regulatory T-cell through inducing TGFBR2 for immunosuppressive tumor microenvironment [146]. From these studies, YAP1 is considered an attractive therapeutic target.

Verteporfin, a porphyrin derivative, is a photosensitizer used in the photodynamic therapy of macular degeneration [20]. Among several drugs, such as dasatinib and statins, that are capable of blocking YAP1 activities [143,147], verteporfin is known as the most effective suppressor of YAP1 activity and widely used in preclinical research [148,149]. Combination treatment of verteporfin and erlotinib suppressed the tumorigenic properties of erlotinib-resistant NSCLC and bladder cancer cells [21,51]. Moreover, the addition of verteporfin to gefitinib decreased the viability of gefitinib-resistant NSCLC and bladder cancer cells [21,52]. Although several ongoing clinical trials are now evaluating the efficacy of verteporfin as a photosensitizer in different types of cancers, none have declared the intent to evaluate YAP1-inhibitory effects.

Our group recently showed that YAP1 is overexpressed in basal-type bladder cancer that has abundant stem cell phenotype [21]. In this study, high YAP1 expression was associated with increased levels of mesenchymal markers, and YAP1 induced CSC properties, such as sphere-forming and self-renewal abilities, invasiveness and drug resistance via induction of SOX2. We also demonstrated that COX2/PGE2 signaling upregulates SOX2 and that a feedback mechanism regulates SOX2 through these pathways. Moreover, combination treatment with verteporfin and a COX2 inhibitor enhanced chemotherapeutic efficacy through the suppression of CSC properties [21]. Further work will be necessary to determine the potential clinical utility of this combination therapy for targeting CSCs.

We have also demonstrated that YAP1 induces STAT3 phosphorylation in lung adenocarcinoma cells by increasing expression of interleukin-6 (IL-6) (manuscript under review). Both YAP1 and STAT3 have been reported to regulate CSCs and other oncogenic pathways [20,21,79]. Indeed, genetic inhibition of both *YAP1* and *STAT3* decreased the stem cell-like features of lung adenocarcinoma cells, resulting in reduced proliferation and enhanced cisplatin sensitivity. Verteporfin was found to suppress not only YAP1 but also STAT3, a previously reported off-target effect of this compound [150]. Using patient-derived xenograft mice models in which the heterogeneity of the primary human tumor was preserved, we found that simultaneous blockade of YAP1 and STAT3 by verteporfin suppressed CSC activity and tumor growth when combined with chemotherapy. Therefore, verteporfin could be an ideal drug for targeting CSC properties by suppressing both YAP1 and STAT3 (Figure 3).

As noted above, verteporfin has already been used to treat of macular degeneration and known to cause minor side effects, such as blurred vision and headache with rare systemic adverse events, making the path to the clinic easier for this drug than for novel YAP1 and STAT3 inhibitors with unknown safety profiles. Although the available evidence suggests that the combination of conventional therapy plus a YAP1 inhibitor holds great promise for the suppression of CSC expansion, we do not yet know the extent to which the heterogeneity of CSCs will be a confounding factor. Further work will be necessary to understand how different classes of CSCs could be targeted.

## 4. Conclusions

In this review, we have discussed some of the potential mechanisms for targeting CSCs, the therapeutic challenges presented in various cancers, and the development of YAP1-targeting strategies. Considering the complexity and diversity of CSCs and non-CSCs, it will be extremely challenging to completely eradicate all cancer cells. However, the addition of CSC-targeting agents to the repertoire of conventional chemotherapeutic drugs represents a promising strategy to induce synergistic effects against both CSCs and non-CSCs.

## Figures and Tables

**Figure 1 cancers-11-00732-f001:**
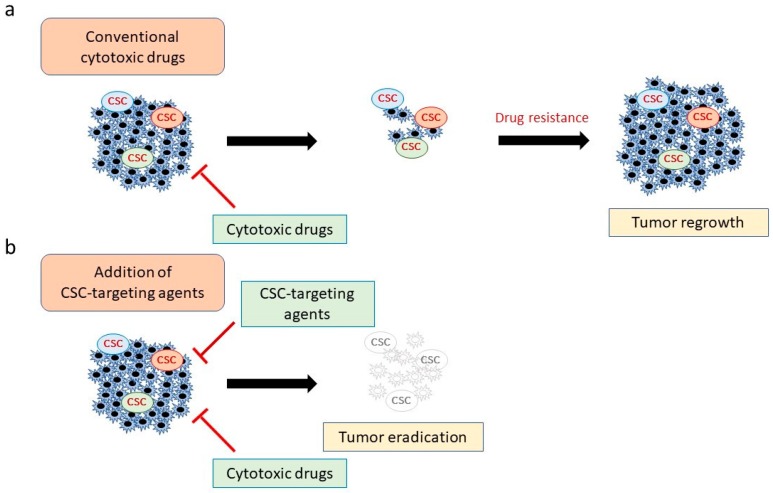
Combining conventional cytotoxic drugs with cancer stem cell (CSC)-targeting agents. (**a**) Although chemotherapeutic and molecular-targeted drugs can attack most cancer cells, CSCs can evade these agents, leading to tumor regrowth. (**b**) Combination therapy with CSC-targeting agents and conventional drugs is predicted to be more effective because it eliminates both CSCs and non-CSC tumor cells.

**Figure 2 cancers-11-00732-f002:**
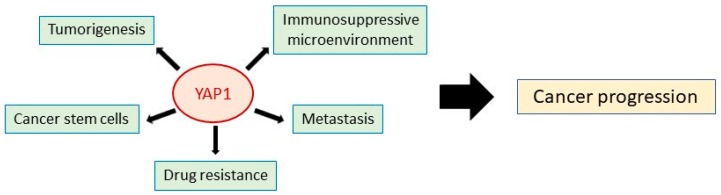
Tumor-promoting roles of YAP1. YAP1 contributes to cancer progression from multiple aspects, such as tumorigenesis, metastasis, malignant stemness, and immunosuppressive microenvironment.

**Figure 3 cancers-11-00732-f003:**
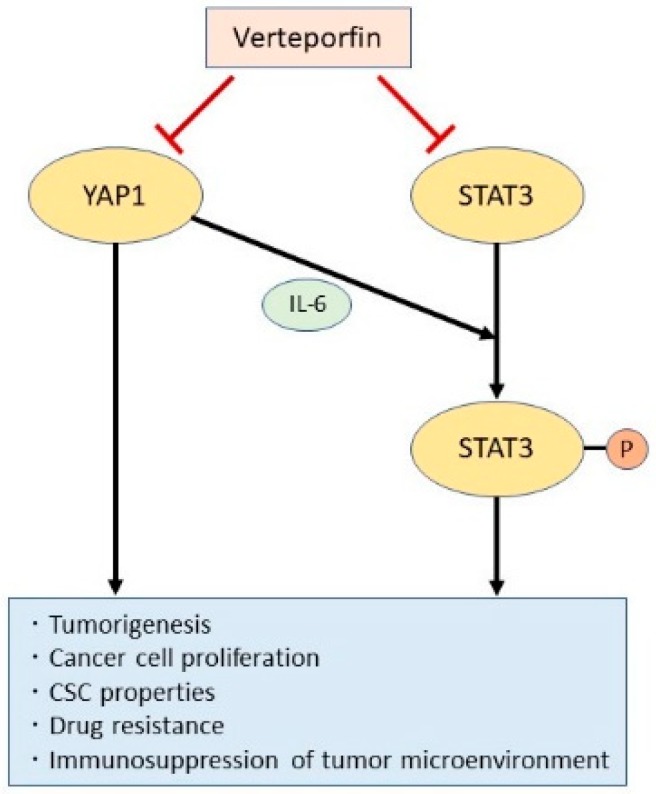
YAP1 and STAT3; Two oncogenic pathways that promote cancer stemness. YAP1 and STAT3 are independently involved in oncogenic signaling to promote cancer stem cell (CSC) properties. However, YAP1 also promotes IL-6-induced STAT3 phosphorylation and activation. The porphyrin derivative verteporfin inhibits both the YAP1 and STAT3 pathways and may thus be an efficient suppressor of CSC properties.

**Table 1 cancers-11-00732-t001:** Therapeutic attempts to target cancer stem cell (CSC).

Target	Cancer Type	Inhibitor	Result	Reference
ALDH1	Breast	Histone deacetylase inhibitor	CD44^+^CD24^−/low^ cell population was decreased and stemness markers were downregulated.	[27,28]
	NSCLC	Disulfiram	Combination of disulfiram and copper downregulated stemness-related genes.	[29]
			When combined with diethylamino-benzaldehyde resensitized resistant cells to cisplatin.	[30]
			A phase II trial showed prolonged survival when disulfiram was combined with cisplatin and vinorelbine.	[31]
	Ovary	Solanum incanum extract	Notch1 and FoxM1 were downregulated, which resulted in increased chemotherapeutic sensitivities.	[32]
CD44	Breast	Anti CD44 antibody	Nanoparticles with CD44 antibody and gemcitabine specifically targeted CD44^+^ cells.	[33]
CD133	Ovary	Anti CD133 antibody-toxin conjugate	Cellular growth was inhibited and tumor progression was suppressed in a mouse model.	[34]
Hedgehog	Bladder	Cyclopamine	Tumor formation was suppressed via inhibition of GALNT1 that mediates SHH signaling.	[35]
	Lung	GDC-0449	Stemness-related features were suppressed in both NSCLC and small-cell lung cancer cells.	[36]
KLF5	Breast	Metformin	CSC growth was inhibited through suppression of *NANOG* and *FGF-BP1* (downstream targets of KLF5).	[37,38]
Notch2 and Notch3	Various cancers Small-cell lung	Tarextumab	Tumorigenesis and cellular growth were suppressed and chemotherapeutic efficacy was increased.	[39]
			A phase Ib trial showed good tolerability and anti-tumor effect.	[40]
PI3K/AKT	Bladder	Myrtucommulone-A	Several stem cell markers were downregulated and stemness-related features were attenuated.	[41]
	Bladder	Motesanib	Survival-related genes in the PI3K/AKT pathway were decreased and cisplatin sensitivity was enhanced.	[42]
STAT3	Breast	STAT3 inhibitor VII	Combination of STAT3 inhibitor and carboplatin abrogated carboplatin-induced ALDH^+^ cell enrichment.	[15]
	Colon	Napabucasin	*c-Myc*, *Nanog* and *Sox2*, were downregulated, which attenuated metastasis in a mouse model.	[43]
			Napabucasin showed prolonged survival in phosphorylated STAT3 positive patients.	[44]
	Pancreas	Napabucasin	Cancer relapse and metastasis were blocked in mice.	[43]
	NSCLC	OPB-51602	A phase I trial suggested that NSCLC patients were likely to obtain better response.	[45]
Wnt/β-catenin	Breast	Pyrvinium pamoate	CD44^+^CD24^−/low^ and ALDH^+^ cells were suppressed by downregulating *NANOG*, *OCT4*, and *SOX2*.	[46]
	Breast	Resveratol	Resveratol, which suppressed Wnt/β-catenin pathway, inhibited CSCs and induced autophagy.	[47]
	Ovary	Imatinib	CSC activity was suppressed when combined with platinum chemotherapy.	[48]
			Phase II clinical trials had only a modest impact on the prognosis of ovarian cancer patients.	[49,50]
YAP1	Bladder	Verteporfin	Combination of YAP1 and COX2 inhibitors with chemotherapy attenuated CSC properties and enhanced chemotherapy response.	[21]
	NSCLC Bladder	Verteporfin	Verteporfin attenuated the resistance to EGFR inhibitors.	[21,51,52]

NSCLC: non-small cell lung carcinoma; SHH: sonic hedgehog signaling molecule.

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
