# Peer review of "Targeting Cancer Stem Cells: A Strategy for Effective Eradication of Cancer"

_cancers, 2019, doi:10.3390/cancers11050732_

Round 1

Reviewer 1 Report

In this manuscript, authors claimed that a strategy for effective therapy via the targeting the cancer stem cells. The manuscript consists of 4 part; 1) introduction, 2) CSC-targeting therapies, 3) targeting CSCs via YAP1, 4) conclusions. Especially, authors focused on YAP1-targeting strategies in the introduction part (Here we review current knowledge of some ~.), and in the part3 (targeting CSCs via YAP1) explained the strategy and role of YAP1 for cancer treatment. However, this manuscript is not focused on YAP1 and the strategy to eliminate the CSCs in cancer type, such as lung, breast, head and neck, bladder, ovarian and colon cancer. Although authors demonstrated the CSC-specific targets and regulators in part2 (CSC-targeting therapies), it is scattered.

Major concern

Therefore, this manuscript required the summarize using a table (including the cancer type, target, references, inhibitor, mode of action, etc) to show the strategies for treatment and CSC-specific targets according to cancer in the part 3.

In this manuscript, authors described that YAP1 is a rising target to develop CSC-targeting therapeutic approaches, and YAP1 is a main effector of Hippo pathway, contribute to drug resistance, and regulate the malignant cell stemness. YAP1 play critical roles in CSCs of several solid tumors. However, the explain for YAP1 also confused and too short. Authors have to add the more detailed explain of YAP1 including the summarized figure.

In the part 3, authors explained the general information about the CSCs together with figure1. This part is to explain the targeting strategies and roles of YAP1 in CSCs. Thereby, the first paragraph (from “recently, several attempts~” to in normal stem cells.) and figure1 must transfer to the first paragraph in the part2.   

Author Response

Comments from Reviewer 1
In this manuscript, authors claimed that a strategy for effective therapy via the targeting the cancer stem cells. The manuscript consists of 4 part; 1) introduction, 2) CSC-targeting therapies, 3) targeting CSCs via YAP1, 4) conclusions. Especially, authors focused on YAP1-targeting strategies in the introduction part (Here we review current knowledge of some ~.), and in the part3 (targeting CSCs via YAP1) explained the strategy and role of YAP1 for cancer treatment. However, this manuscript is not focused on YAP1 and the strategy to eliminate the CSCs in cancer type, such as lung, breast, head and neck, bladder, ovarian and colon cancer. Although authors demonstrated the CSC-specific targets and regulators in part2 (CSC-targeting therapies), it is scattered.

We thank the reviewer for giving us these thoughtful comments.

We now revised the article by considering all the comments/suggestions raised by the reviewer.

Major concern

Therefore, this manuscript required the summarize using a table (including the cancer type, target, references, inhibitor, mode of action, etc) to show the strategies for treatment and CSC-specific targets according to cancer in the part 3.

Response: We added a table (Table 1) that summarized the strategies to target CSC-specific molecules or pathways.

In this manuscript, authors described that YAP1 is a rising target to develop CSC-targeting therapeutic approaches, and YAP1 is a main effector of Hippo pathway, contribute to drug resistance, and regulate the malignant cell stemness. YAP1 play critical roles in CSCs of several solid tumors. However, the explain for YAP1 also confused and too short. Authors have to add the more detailed explain of YAP1 including the summarized figure.

Response: We appreciate the reviewer’s well thought comment. YAP1 has variety of functions to promote cancer progression. We revised the part that described YAP1 roles (line 332-387) and added Figure 2 (line 388).

In the part 3, authors explained the general information about the CSCs together with figure1. This part is to explain the targeting strategies and roles of YAP1 in CSCs. Thereby, the first paragraph (from “recently, several attempts~” to in normal stem cells.) and figure1 must transfer to the first paragraph in the part2.

Response: We agree with the reviewer’s comment. Figure 1 has been transferred to the part 2 (line 77).

In addition, we rewrote the part of line 174-175 by citing the new article.

Reviewer 2 Report

Overall comments:

The review article entitled “Targeting Cancer Stem Cells: A Strategy for Effective Eradication of Cancer” introduced the therapeutic strategies for targeting CSCs as well as the ongoing clinical trials of pathway inhibitors in breast, colorectal, lung, bladder, and head and neck cancer. Also they further attempted to summarize the potential of YAP1 as a promising CSC target and proposed Verteporfin as a future combinatory drug to enhance the cellular sensitivity to non-targeted conventional therapy. Although the manuscript includes remarkably comprehensive and is richly referenced, the format of the manuscript is not satisfactory. The Sub-topics are just divided according to the cancer types. The way CSC targets are presented is not well-organized and the number is very limited. In particular, authors proposed YAP1 as a future promising target for CSCs, however they focused on only STAT3 signaling as its regulatory mechanism. Therefore, a substantial modification is required.

Specific comments:

1. Since the authors emphasized the importance of CSC-targeting drugs regarding their potential application for enhancing therapeutic response, the potential CSC targets have to be introduced regarding their relevance in CSC sensitization to chemotherapy in every main category.

2. A table summarizing the CSC-related targets and their specific inhibitors have to be provided with the current status of clinical investigation to clear the conclusion of this manuscript.

3. The molecular mechanisms of YAP1 discussed in this manuscript is biased to STAT3 signaling. More references should be added to describe the complicated CSC signaling networks of YAP1. For instance, YAP1 can trigger Wnt signaling by forming complex with β-catenin1), and it also directly up-regulates the transcription of SOX92). There are plenty of references about the YAP1 mechanism.

4. If it is a general idea that YAP1 is not required for normal tissue homeostasis, more evidences have to be provided besides ref# 133. At current reviewing, a plenty of evidences suggest that YAP1 is involved in adult stem cell regulation3),4),5),6). Thus this sentence have to be removed or richly referenced.

5. In addition, authors stated that YAP1 is important because it is overexpressed in various types of cancer and because it is activated by CSC-associated pathways and CSC-associated lncRNAs (line#320-334). However, these are insufficient to establish rationale why the authors espacially focused on YAP1 as a potential CSC target. Thus more references have to be incorporated such as the diverse mechanisms leading to YAP1 activation in the context of CSC metabolism and microenvironment to emphasize the importance of YAP1 and to attract broad readership7).

6. A table summarizing the YAP1 inhibitors currently on clinical or preclinical investigation have to be provided for a concise synopsis of the development of YAP1-targeting strategies.

Minor concerns:

7. The manuscript seems to have a mistake which critically changes the whole meaning of sentence.

-At line #337: “For example, in the absence of Wnt signaling, YAP1 is restricted to the β-catenin destruction complex, but initiation of Wnt signaling induces their release and translocation to the nucleus, where they upregulate transcription of β-catenin and YAP1 target genes”

-I think that “inhibition” might be a mistake. “activation” may be adequate for this sentence.

8. The meaning of the sentences is vague; thus it has to be rephrased or eliminated.

-At line #359: “YAP1 and STAT3 can also independently regulate CSCs and other oncogenic pathways [20,21,58].”

- The reference #20, 21, 58 show the mechanism of YAP1 or STAT3 in regulating cancer stemness. They do not present evidences that YAP1 can regulate stemness in a STAT3-indepdent manner.

9. Some sentences have grammar issues.

-At line 18: “CSCs are often drug resistance

I think that the sentence sounds strange and “resistant” may be more appropriate.

References:

Deng F, et al. YAP triggers the Wnt/β-catenin signalling pathway and promotes enterocyte self-renewal, regeneration and tumorigenesis after DSS-induced injury. Cell Death Dis. 2018 Feb 2;9(2):153.

Wang L, et al. Unbalanced YAP–SOX9 circuit drives stemness and malignant progression in esophageal squamous cell carcinoma. Oncogene. 2019 Mar;38(12):2042-2055.

Elbediwy A, et al. Integrin signalling regulates YAP and TAZ to control skin homeostasis. Development. 2016 May 15;143(10):1674-87.

Lu L, et al. Hippo pathway coactivators Yap and Taz are required to coordinate mammalian liver regeneration. Exp Mol Med. 2018 Jan 5;50(1):e423.

Mao B, et al. Hippo signaling in stress response and homeostasis maintenance. Acta Biochim Biophys Sin (Shanghai). 2015 Jan;47(1):2-9

Panciera T, et al. Mechanobiology of YAP and TAZ in physiology and disease. Nat Rev Mol Cell Biol. 2017 Dec;18(12):758-770

Zanconato F, et al. YAP/TAZ at the Roots of Cancer. Cancer Cell. 2016 Jun 13;29(6):783-803.

Author Response

Comments from Reviewer 2
The review article entitled “Targeting Cancer Stem Cells: A Strategy for Effective Eradication of Cancer” introduced the therapeutic strategies for targeting CSCs as well as the ongoing clinical trials of pathway inhibitors in breast, colorectal, lung, bladder, and head and neck cancer. Also they further attempted to summarize the potential of YAP1 as a promising CSC target and proposed Verteporfin as a future combinatory drug to enhance the cellular sensitivity to non-targeted conventional therapy. Although the manuscript includes remarkably comprehensive and is richly referenced, the format of the manuscript is not satisfactory. The Sub-topics are just divided according to the cancer types. The way CSC targets are presented is not well-organized and the number is very limited. In particular, authors proposed YAP1 as a future promising target for CSCs, however they focused on only STAT3 signaling as its regulatory mechanism. Therefore, a substantial modification is required.

We appreciate the reviewer for reading the details of our paper and giving us several thoughtful comments. We now revised the article by considering all the comments/suggestions raised by the reviewer.

Specific comments:

1. Since the authors emphasized the importance of CSC-targeting drugs regarding their potential application for enhancing therapeutic response, the potential CSC targets have to be introduced regarding their relevance in CSC sensitization to chemotherapy in every main category.

Response: We thank the reviewer’s appropriate suggestion. We newly made Table 1 to summarize the attempts to target CSCs in many types of cancers.

2. A table summarizing the CSC-related targets and their specific inhibitors have to be provided with the current status of clinical investigation to clear the conclusion of this manuscript.

Response: We made Table 1 that summarized the CSC targeting therapies according to the reviewer’s suggestion.

3. The molecular mechanisms of YAP1 discussed in this manuscript is biased to STAT3 signaling. More references should be added to describe the complicated CSC signaling networks of YAP1. For instance, YAP1 can trigger Wnt signaling by forming complex with β-catenin1), and it also directly up-regulates the transcription of SOX92). There are plenty of references about the YAP1 mechanism.

Response: We added more references to describe YAP1 mechanism (line 332-386) and made Figure 2 to show the YAP1’s multifunctional characteristics (line 388).

4. If it is a general idea that YAP1 is not required for normal tissue homeostasis, more evidences have to be provided besides ref# 133. At current reviewing, a plenty of evidences suggest that YAP1 is involved in adult stem cell regulation3),4),5),6). Thus this sentence have to be removed or richly referenced.

Response: We sincerely thank the reviewer for collecting our misunderstanding. We omitted the sentence from the manuscript (line 338).

5. In addition, authors stated that YAP1 is important because it is overexpressed in various types of cancer and because it is activated by CSC-associated pathways and CSC-associated lncRNAs (line#320-334). However, these are insufficient to establish rationale why the authors especially focused on YAP1 as a potential CSC target. Thus more references have to be incorporated such as the diverse mechanisms leading to YAP1 activation in the context of CSC metabolism and microenvironment to emphasize the importance of YAP1 and to attract broad readership7).

Response: We appreciate the reviewer’s the thoughtful comments. We added more references and noted that YAP1 in the cells that composed tumor microenvironment also contribute to cancer progression (line 380-387).

6. A table summarizing the YAP1 inhibitors currently on clinical or preclinical investigation have to be provided for a concise synopsis of the development of YAP1-targeting strategies.

Response: We agree with the reviewer’s comment. However, as far as we know, only several inhibitors, such as dasatinib and statins, have been reported to inhibit YAP1. In most preclinical studies, verteporfin has been used to target YAP1. This is why we added a sentence instead of a table (line 393-394).

Minor concerns:

7. The manuscript seems to have a mistake which critically changes the whole meaning of sentence.

-At line #337: “For example, in the absence of Wnt signaling, YAP1 is restricted to the β-catenin destruction complex, but initiation of Wnt signaling induces their release and translocation to the nucleus, where they upregulate transcription of β-catenin and YAP1 target genes”

-I think that “inhibition” might be a mistake. “activation” may be adequate for this sentence.

Response: We thank the reviewer for collecting our mistake. We changed ‘inhibition’ to ‘activation’ (line 352).

8. The meaning of the sentences is vague; thus it has to be rephrased or eliminated.

-At line #359: “YAP1 and STAT3 can also independently regulate CSCs and other oncogenic pathways [20,21,58].”

- The reference #20, 21, 58 show the mechanism of YAP1 or STAT3 in regulating cancer stemness. They do not present evidences that YAP1 can regulate stemness in a STAT3-indepdent manner.

Response: We thank the reviewer for pointing out the important thing. We changed this sentence as follows. “Both YAP1 and STAT3 have been reported to regulate CSCs and other oncogenic pathways” (line 411-412).

9. Some sentences have grammar issues.

-At line 18: “CSCs are often drug resistance

I think that the sentence sounds strange and “resistant” may be more appropriate.

Response: That is true. We changed the word ‘resistance’ to ‘resistant’ (line 18).

In addition, we rewrote the part of line 174-175 by citing the new article.

Round 2

Reviewer 1 Report

.

Reviewer 2 Report

My concerns have been well addressed. The review article entitled “Targeting cancer stem cells: A strategy for effective eradication of cancer” introduced the potential therapeutic targets for cancer stem cells (CSCs) and further attempted to summarize the potential combinatory drugs with molecular mechanisms. The structure of the manuscript is clear and the conclusion made from this review may be of interest in the relevant field.